# Multimode polariton effects on molecular energy transport and spectral fluctuations

Raphael F. Ribeiro [1] ✉

Despite the potential paradigm breaking capability of microcavities to control chemical processes, the extent to which photonic devices change properties of molecular materials is still unclear, in part due to challenges in modeling hybrid light-matter excitations delocalized over many length scales. We overcome these challenges for a photonic wire under strong coupling with a molecular ensemble. Our simulations provide a detailed picture of the effect of photonic wires on spectral and transport properties of a disordered molecular material. We find stronger changes to the probed molecular observables when the cavity is redshifted relative to the molecules and energetic disorder is weak. These trends are expected to hold also in higher-dimensional cavities, but are not captured with theories that only include a single cavity-mode. Therefore, our results raise important issues for future experiments and model building focused on unraveling new ways to manipulate chemistry with optical cavities.

---

[1] Department of Chemistry and Cherry Emerson Center for Scientific Computation, Emory University, Atlanta, GA, USA. ✉email: raphael.ribeiro@emory.edu

Strong light–matter interactions hosted by nanostructures and optical microcavities can induce significant and qualitative changes to chemical processes[1–3] including photoconductivity[4,5], energy transport[6–8], and optical nonlinearities[9–11]. Much of the observed phenomenology stems from the hybridization of the collective material polarization and the resonances of an optical cavity, which leads to the formation of delocalized polariton modes when the energy exchange between the collective molecular excitations and the photonic structure is faster than the dissipative processes acting on each subsystem[9,12–15]. Polaritonic states are always accompanied by molecular states weakly coupled to light[16]. The latter are also sometimes called "dark states", as the optical response of strongly coupled molecular ensembles is dominated by polaritonic excitations. The weakly coupled states form a reservoir containing the vast majority of the states of the system with significant molecular character[16–18], and therefore, they play an essential role in equilibrium and non-equilibrium molecular phenomena in optical cavities[13,19–23].

Polaritons dominate the optical response of a strongly coupled device, but the reservoir states are much more numerous[16,18]. Therefore, it is puzzling that significant changes in thermal reaction rates and branching ratios of some organic systems[24–29] were observed under conditions of infrared collective strong coupling. The dominance of molecular reservoir modes over the polaritonic[18,30,31] suggests there is no simple explanation of the cavity effect on thermal reactions based on transition-state theory[32–34]. This notion has motivated the hypothesis that cavity-induced changes to chemical reactions originate from dynamical effects of the electromagnetic environment on intramolecular dynamics[35–37] (see also[38–41], for theoretical treatments of strong light–matter interaction effects on chemical reaction dynamics when a single molecule is strongly coupled to a cavity-photon mode and dark states are absent).

Given the complexity of polaritonic systems with states delocalized over length scales of the order of the optical wavelength, much of what is known about collective strong coupling relies on quantum mechanical simulations of effective models[42,43] where the cavity is modeled as a single bosonic mode and the molecular system has permutational symmetry, i.e., it consists of an ensemble of identical $N_M \gg 1$ two-level system with equal transition energy and dipole moment[13,14,18]. In these models, hybrid light–matter excitations (lower and upper polaritons, LP and UP, respectively) extending over the entire system emerge from the interaction of the homogeneous cavity field with the molecular bright mode corresponding to the totally symmetric combination of molecular states where a single molecule is excited. The other $N_M - 1$ molecular modes are degenerate and correspond to the dark reservoir discussed above. Permutationally invariant models deviating from this Tavis-Cummings (TC) picture via the introduction of exciton-phonon interactions[18,44–46] have also been thoroughly investigated.

The effects of material imperfections on polaritonic and dark states were probed in early work by Houdre et al.[17] who showed that the presence of energetic and structural disorder lead to weak photonic intensity borrowing by the reservoir states, but only minor changes to the TC picture when the collective light–matter interaction $\Omega_R/2$ (Rabi splitting) is much greater than the mean fluctuation $\sigma$ of the material transition energies. Recently, Scholes revisited the polariton coherence protection in single-mode (0D) cavities[47], whereas Botzung et al.[48] provided quantitative localization properties of the reservoir states of an energetically disordered emitter ensemble under strong coupling to a single spatially homogeneous cavity mode. Both studies showed in agreement with ref. [17] that polariton coherence is largely unaffected by energetic disorder weaker than the collective

light–matter coupling, but also noted that dark modes inherit spatial delocalization (see also refs. [19,49] for identification of signatures of dark-state delocalization in energy transport simulations, and ref. [50] where properties of polaritonic and dark excitations in a leaky single-mode cavity are given).

However, the majority of photonic materials employed in polariton chemistry research are multimode Fabry-Perot (FP) or plasmonic cavities with a continuous spectrum[51,52]. For example, in an FP cavity, each electromagnetic mode is characterized by its polarization (TE or TM), cavity order $m = 1, 2, 3...$ (equivalent to the longitudinal wave-vector $k_z$), and the essentially continuous in-plane wave-vector $\mathbf{q} = (q_x, q_y)$. In systems such as FP cavities, the disorder is known to severely restrict polariton space-time coherence[16,53] and may lead to weak or strong Anderson localization as well as diffusive and ballistic transport depending on the initial wave-packet, disorder strength, cavity geometric parameters, and magnitude of light–matter interactions[54]. Numerical simulations have shown that polariton wave functions can be localized over different length scales depending on their mean wave-vector[55–57] (see ref., [58] for a recent study of multimode-cavity effects on polariton relaxation).

Despite their prominence, much less attention has been paid to multimode strong coupling effects on the reservoir states of structurally and energetically disordered molecular ensembles. These states form the majority of the excitations with significant molecular character, and therefore, they largely determine the equilibrium and transport properties of a molecular subsystem under collective strong coupling with an optical cavity.

Here, we employ numerical simulations of the microscopic states of a multimode photonic wire under strong coupling with a disordered molecular ensemble to investigate the influence of the cavity on the local molecular density of states and the exciton return probability. These quantities essentially reflect the properties of the reservoir modes (since they are much more numerous than the polaritonic) and allow us to quantitatively probe the effects of multimode optical cavities on the molecular ensemble.

Our results have significant implications for future model building in polariton chemistry since they provide detailed illustrations of qualitative shortcomings of single-mode representations of multimode devices. We also suggest practical principles to enhance cavity effects on molecular properties likely holding for generic systems, e.g., we find that for systems with equal collective light–matter interaction, polariton effects on the molecular ensemble are largest when the cavity is redshifted, the distance between molecules is largest and energetic disorder is weak. Wave function localization theory[53,54,59] and simple spectral overlap arguments explain these facts which provide another piece to the puzzle[60] of the optical cavity effect on chemical reactions since some experimental observations suggest that the influence of photonic materials on chemical reactions is greater when the cavity-matter detuning vanishes[61].

## Results and discussion

**Microscopic model**. We model an optical microcavity with $O(\mu m)$ longitudinal and lateral confinement lengths $L_z$ and $L_y$ along the $z$ and $y$ axes, respectively[62]. The length of the long axis is $L \gg L_z, L_y$. The molecules are distributed homogeneously along $x$ and have equal $y$, $z$ coordinates (see Fig. 1). Ideal reflective surfaces confine the EM field along $z$ and $y$, whereas periodic boundary conditions are assumed along $x$. We include a single polarization of the EM field parallel to the direction of each molecule's transition dipole moment. The bare cavity modes have frequency $\omega = (c/\sqrt{\epsilon})k$, where $c$ is the speed of light, $\epsilon$ is the static dielectric constant of the intracavity medium and $k$ is the magnitude of

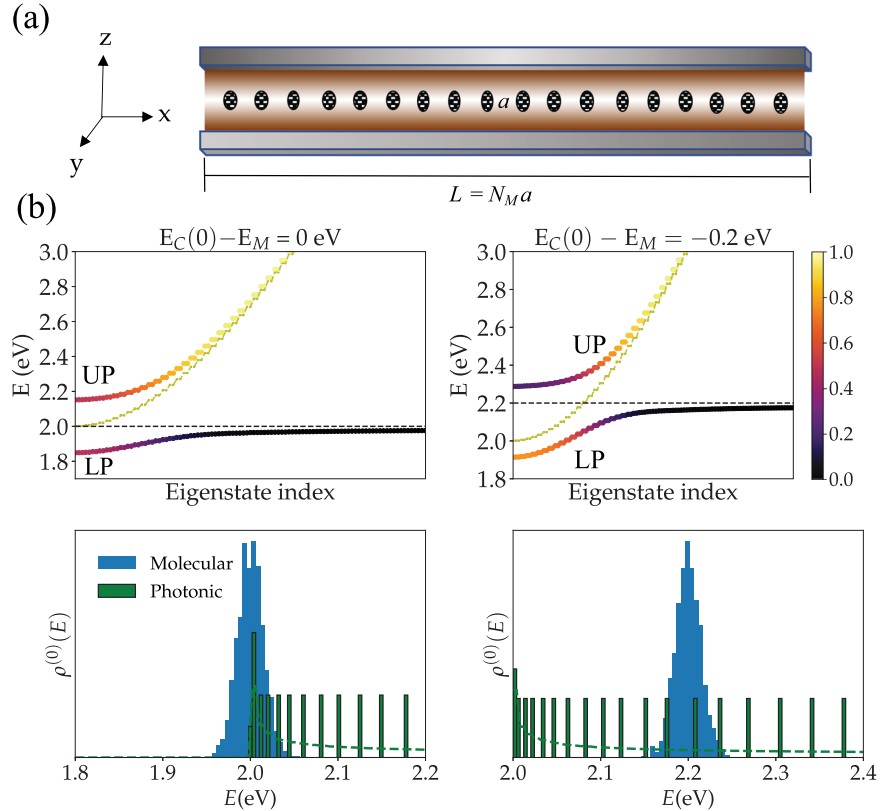

**Fig. 1 Schematic representation of photonic wire model for multimode polariton chemistry. a** Section of the photonic wire model employed in this work. Molecules (ovals) extend homogeneously (in average) along $x$ with constant $z$ and $y$. The distance between the $N_M$ molecules is in average $a$. **b** Subset of the energy spectrum (obtained from exact diagonalization of a single realization of the Hamiltonian of a system with 1001 molecules and cavity modes, $\Omega_R = 0.3$eV, $\sigma/\Omega_R = 0.05$, and other parameters as listed in "Thermodynamic limit convergence") of the empty cavity (dashed yellow), lower polariton (LP), and upper polariton modes (UP) are shown as a function of eigenstate index $i = 1, 2,...,1001$, with $E_1 < E_2 <...$ for cavities with $q = 0$ (top left) and $q > 0$ (top right) modes on-resonance with the disordered molecular ensemble (mean excitation energy given by the horizontal dashed line). Only the first 100 modes of the LP and UP are shown. Each point is colored according to the photonic content of the corresponding state, where darkest corresponds to zero and brightest to 100%. Thus, bright points correspond to visible peaks in transmission spectra. The LP band includes the weakly coupled molecular reservoir modes ("dark states") as is clear from the color of the corresponding points. The bottom left and bottom right shows the (single realization) bare molecule DOS, and rescaled (for the sake of clarity, as the exact number is too small compared to the molecular) discretized photonic DOS for representative zero and red-detuned cavities (in the $N_C \rightarrow \infty$ limit, the DOS becomes continuous and a singularity arises at $E_C(0)$). Note that in the zero-detuning case, the subset of molecules with the lowest energies is off-resonant with all cavity modes and therefore the effects of the confined field on such molecules is smaller. Dashed green curves in each of the bottom figures correspond to smooth interpolations of the cavity density of states from the computed discrete mode distributions presented as histograms.

the three-dimensional wave-vector $\mathbf{k} = (2\pi m_x/L, n_y\pi/L_y, n_z\pi/L_z)$, where $m_x \in \mathbb{Z}$, and $n_y$ and $n_z$ are positive integers. The parameters $L$, $L_y$, $L_z$, and $\epsilon$ are chosen such that the vast majority of the molecules are only resonant with cavity modes in their lowest-energy band ($n_z = n_y = 1$). Therefore, we ignore all other bands, and only include photons with $\mathbf{k} = (2\pi m/L, \pi/L_y, \pi/L_z)$, $m \in \mathbb{Z}$. From now on, we label the cavity modes by $q \equiv k_x$, omit any reference to $k_y$ and $k_z$ and identify $q$ as the light wave-vector degree of freedom. The empty cavity Hamiltonian is given by:

$$H_L = \sum_q \hbar\omega_q a_q^\dagger a_q, \quad \omega_q = \frac{c}{\sqrt{\epsilon}}\sqrt{q_0^2 + q^2}, \tag{1}$$

where $q_0 = \sqrt{(\pi/L_z)^2 + (\pi/L_y)^2}$. The $q = 0$ cavity mode has lowest energy $E_C(0) = \hbar c q_0/\sqrt{\epsilon}$.

The molecular ensemble is represented by a set of $N_M = L/a$ two-level systems with transition frequencies sampled from a Gaussian distribution (representing low-frequency fluctuations of the solvent environment around each molecule) with mean $E_M$ and variance $\sigma^2$, so the transition energy for the $i$th molecule is $E_i = E_M + \sigma_i$, where $\langle\sigma_i\rangle_d = 0$ and $\langle\sigma_i\sigma_j\rangle_d = \sigma^2\delta_{ij}$. We also

include structural disorder in the form of random deviations of the molecular center of mass positions relative to a perfect crystal arrangement with lattice spacing $a$, and by allowing the single-molecule transition dipole moment to deviate weakly from the mean value $\mu_0 > 0$. The $i$th molecule position is $x_i = (i-1)a + \Delta x_i$ (mod $N_M a$), where $\Delta x_i = f_i a$, and $f_i$ is sampled from a uniform distribution over $[-f, f] \subset \mathbb{R}$. The parameter $f$ controls the maximal $(1 + 2f)a$ and minimal $(1 - 2f)a$ distances between neighboring molecules, respectively. The transition dipole moment of the $i$th molecule is given by the random variable $\mu_i$ sampled from a normal distribution with mean $\mu_0 > 0$ and variance $\sigma_\mu^2$. The structural disorder is typically neglected in numerical treatments of disorder effects on polaritons, but is included here since when $\Omega_R \gg \sigma$, polariton localization at low energies may be primarily driven by fluctuations in the molecular position and dipole moments (relative to a perfectly ordered system)[53,54]. Nevertheless, for the studied model, except for infinitesimal values of $\sigma$ and simultaneous $\sigma_\mu/\mu_0$ close to or greater than 1, energetic disorder plays a more important role, and therefore, we take $\sigma_\mu$ and $f$ to be constant throughout this work. A detailed discussion of energetic and structural disorder effects on molecular observables is given in

Supplementary Note 2 including a quantitative comparison of structural and energetic disorder on the exciton escape probability $\chi_M$ and molecular local density of states entropy variation $\Delta S[\rho_M]$ (see Supplementary Figure 1).

Assuming $a$ is sufficiently large, the Hamiltonian for the bare molecules is:

$$H_M = \sum_{i=1}^{N_M} (E_M + \sigma_i) b_i^+ b_i^-, \qquad (2)$$

where $E_M = \hbar\omega_M$, and $b_i^+ (b_i^-)$ creates (annihilates) an excitation at the $i$th molecule. These operators can be written as $b_i^+ = |1_i\rangle\langle 0|$ and $b_i^- = |0\rangle\langle 1_i|$, where $|0\rangle$ is the state where all molecules and cavity modes are in their ground-state, and $|1_i\rangle$ is the state where only the $i$th molecule is excited. The total Hamiltonian is given by:

$$H = H_L + H_M + H_{LM}, \qquad (3)$$

where $H_{LM}$ contains the light–matter interaction. We employ the Coulomb gauge[63] in the rotating-wave-approximation since we take $\Omega_R/2 < 0.1 E_M$[64]. It follows that,

$$H_{LM} = \sum_{j=1}^{N_M} \sum_q \frac{-i\Omega_R}{2} \sqrt{\frac{\omega_M}{N_M \omega_q}} \frac{\mu_j}{\mu_0} \left( e^{iqx_j} b_j^+ a_q - e^{-iqx_j} a_q^\dagger b_j^- \right) \qquad (4)$$

where $\Omega_R = \mu_0\sqrt{\hbar\omega_0\rho/2\epsilon}$, $\rho = N_M/LS$, and $S = L_y L_z$. We ignored the diamagnetic contribution ($A^2$ term) to $H_{LM}$ since its effects are negligible under the studied conditions[65].

For a given $\Omega_R$ and $a = 1/\rho S$, $N_M$, and $N_C$ are free parameters. Any particular choice is equivalent to imposing low and high-energy cutoffs to the EM field. Specifically, $L = N_M a$ defines the resolution of the cavity in reciprocal space $\Delta q = 2\pi/L$, and $N_C$ defines the maximal cavity-mode energy $E_{max}$. Simulation results are independent of these cutoffs as long as $L$ is larger than the longest coherence length of the system and $E_{max}$ is greater than any relevant energy scale. Alternatively, thermodynamic limit ($N_M, L \to \infty$ with fixed $\rho$) independence of molecular observables to the number of included degrees of freedom can be imposed to obtain a minimal number of molecular and cavity modes. Such computationally optimal number of modes can be strongly dependent on the molecular observable of interest and may also vary significantly with $\Omega_R$ and $\sigma$.

We employ $N_C = N_M$ in our simulations below, since in this case, the thermodynamic limit is reached with a small number of disorder realizations for the observables and range of parameters probed in our studies. A significant fraction of cavity modes is highly off-resonant with the molecular system in every case studied ($N_M \geq 1001$), and we checked that a smaller number of cavity modes $N_C^0$ would suffice to obtain thermodynamic limit predictions. However, as mentioned above, $N_C^0$ depends on various parameters and we leave for future work a detailed analysis of optimal multimode-cavity representations[66] for the study of the effects of photonic devices on molecules.

In later sections, we characterize the dependence of local molecular observables on the energetic disorder, cavity detuning, and mean intermolecular distance (or photonic lattice constant) for fixed $N_M = N_C$ and $\Omega_R$. The study of lattice constant effects with fixed $N_M = N_C$ is a distinctive feature of our work in relationship to refs. [53,56] (who analyzed polariton coherence in the thermodynamic limit with fixed $a$) and is motivated by the following question: are there significant differences in the polariton effects on molecules showing equal $\Omega_R$ but different molecular densities (1/$a$ in our photonic wire)? Note that a change in the lattice constant $a$ to $f \times a$ ($f > 0$) leaves $\Omega_R$ fixed under the mentioned conditions if an only if $\mu_0 \mapsto \sqrt{f}\mu_0$ (as $\Omega_R \propto \mu_0 a^{-1/2}$), as expected since a reduced molecular density requires greater transition dipole moment per

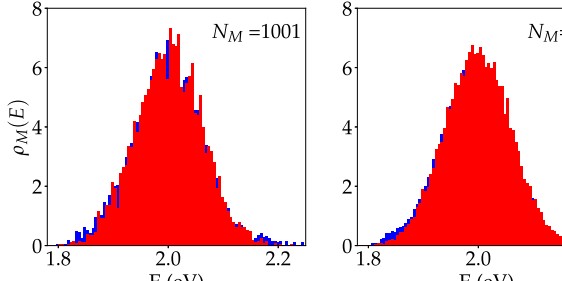

**Fig. 2 Ensemble limit of local molecular density of states in polaritonic wire.** Local molecular density of states obtained from five realizations of a molecular system in free space (red) and under strong coupling (blue) to a photonic wire ($N_C = N_M$) with $E_M - E_C(0) = 0$ (zero detuning), $N_M = 1001$ and 5001, $\Omega_R = 0.3$eV, $\sigma/\Omega_R = 0.2$, $\sigma_\mu = 0.05\mu_0$, and $f = 0.1$ in both cases. Other parameters are listed in the text (See Thermodynamic limit convergence).

molecule to preserve a given Rabi splitting. Nonetheless, a potential unintended consequence of changing $a$ with fixed $N_C$ in cavity lattice models is that the photon density of states $\rho_C(\omega) = \sum_q \delta(\omega - \omega_q)$ is also modified. The thermodynamic limit, in fact, $\rho_C(\omega) \to [L/(2\pi)]\int_{\mathbb{R}} dq\, \delta(\omega - \omega_q)$ is proportional to $L$, and only $\rho_C(\omega)/L$ is independent of $L$. In Fig. 2, we show that the effects of optical microcavities in the molecular ensemble measured in our study are essentially independent of $L$ as long as a sufficiently large number of modes is introduced in the theory. This suggests that the change in photon density of states that result from varying $a$ (with fixed $N_C = N_M$ and $\Omega_R$) is immaterial to our conclusions. In Supplementary Note 1, we provide further discussion and numerical evidence that further supports these points (see Supplementary Tables I and II).

**Observables.** Let the eigenstates and eigenvalues of $H$ be denoted by $\psi$ and $E_\psi$, respectively. The ensemble-averaged molecular local density of states (LDOS) gives the conditional probability that an excited molecule will be detected with energy $E$:

$$\rho_M(E) = \frac{1}{N_M} \sum_{n=1}^{N_M} \langle 1_n | \delta(\hat{H} - E) | 1_n \rangle$$
$$= \sum_\psi \langle P_{n\psi} \rangle \delta(E_\psi - E), \qquad (5)$$

$P_{n\psi} = |\langle 1_n | \psi \rangle|^2$ and $\langle P_{n\psi} \rangle = \sum_{n=1}^{N_M} |\langle 1_n | \psi \rangle|^2 / N_M$ is the average probability to find a specific molecule excited when the system is in the eigenstate $|\psi\rangle$. In the absence of light–matter interactions, the molecular LDOS $\rho_M^{(0)}(E)$ is a Gaussian distribution centered at $E_M$ and width $\sigma$. We quantify the photonic effect on $\rho_M(E)$ by evaluating the cavity-induced change in its Shannon entropy

$$\Delta S = S[\rho_M] - S[\rho_M^{(0)}], \qquad (6)$$

$$S[\rho_M] = -\int_0^\infty dE\, \rho_M(E)[\rho_M(E)]. \qquad (7)$$

$\Delta S[\rho_M]$ allows us to quantify the cavity effect on the molecular ensemble energy fluctuations. Roughly speaking, $\Delta S[\rho_M]$ provides a measure of molecular excited-state delocalization in energy space. Therefore, we expect $\Delta S[\rho_M]$ to be a nondecreasing function of $\Omega_R$ that is greater than or equal to zero, since polaritons have energies separated from the bare molecule excited-states by approximately $\pm \Omega_R/2$ (Fig. 1). However, the reduced bare photonic DOS relative to the molecular at energies where the molecular DOS is maximal suggests weakly coupled reservoir

states dominate the molecular LDOS and the cavity-driven change in $S[\rho_M]$ is expected to be small.

The mean probability that an initially excited molecule at time $t = 0$ will be detected in the same state at $t > 0$ is the time-dependent exciton survival (return) probability $P_M(t) = \sum_{j=1}^{N_M} P_j(t)/N_M$, where

$$P_j(t) = \left| \left\langle 1_n | e^{-iHt/\hbar} | 1_n \right\rangle \right|^2. \tag{8}$$

The exciton return probability $\Pi_M$ is the $t \to \infty$ limit of $P_M(t)$:

$$\Pi_M \equiv \lim_{t \to \infty} P_M(t) = \frac{1}{N_M} \sum_{n=1}^{N_M} \sum_{\psi} P_{n\psi}^2$$
$$= \sum_{\psi} \left\langle P_{n\psi}^2 \right\rangle. \tag{9}$$

The exciton escape probability $\chi_M$ is simply related to $\Pi_M$ via:

$$\chi_M = 1 - \Pi_M. \tag{10}$$

This quantity provides the ensemble-averaged probability that energy initially stored as a localized molecular exciton migrates to a distinct molecule or is converted into a cavity photon after an infinite amount of time.

$\Pi_M$ provides a measure of excited-state delocalization in real-space and coherent energy diffusion efficiency: in systems where all Hamiltonian eigenstates are delocalized $P_{n\psi} \propto 1/N_M$ for all $\psi$ and $n$, and therefore, $\Pi_M$ vanishes in the thermodynamic limit, whereas in noninteracting systems with maximally localized excited-states, each molecule corresponds to a Hamiltonian eigenstate, and therefore $P_{n\psi}(t) = \delta_{n\psi}$ and $\Pi_M = 1$. In our model, the case where $\Pi_M = 1$ corresponds to the molecules outside of the cavity (since we assume direct intermolecular interactions are insignificant), while we find $\Pi_M \to 0$ when energetic disorder vanishes. Dark states are expected to have a much large contribution to $\Pi_M$ than polaritons since the latter inherit greater delocalization from their strong mixing with cavity modes.

Note that for a disordered multimode system there is no unambiguous definition of polariton and dark states since there is no energy gap between the LP band and the reservoir modes which are weakly coupled to the cavity. However, given a definition of polariton and weakly coupled modes, it is possible to decompose $\Pi_M$ and $\chi_M$ into a sum of contributions from polariton and reservoir modes and gauge the sensitivity of polariton and dark-state delocalization to increasing disorder. To analyze our numerical simulations, we choose polaritons to consist of all states where the total photonic content is >15%, but <85%, whereas dark states are all eigenstates with total molecular content >85%. We briefly discuss this choice in the section "Density and energetic disorder dependence".

**Thermodynamic limit convergence.** Before presenting a detailed quantitative study of the photonic effects on the mentioned molecular observables, we first show that with a relatively small number of molecules and modes we obtain robust predictions for cavity-induced changes in molecular properties. We set $L_y = 400$ nm, $L_z = 200$ nm, and dielectric constant $\epsilon = 3$. This gives the lowest-energy cavity mode with $E_C(0) \equiv \hbar\omega_0 = 2.0$ eV. The highest energy wave-vector is $q_{max} \approx \pi/a$ with $a = 10$, 25, and 50 nm. To probe the thermodynamic limit of $\rho_M(E)$ and $\Pi_M$, we take $\Omega_R = 0.3$ eV, $E_M = 2.0$ eV, $\sigma = 0.2\,\Omega_R$, $f = 0.1$, and $\sigma_\mu/\mu_0 = 0.05$. To compute the molecular LDOS we employed 400 bins of width 5 meV spanning the interval [1.5 eV, 3.5 eV].

In Fig. 2, we show $\rho_M(E)$ obtained with five realizations of a system with $N_M = N_C = 1001$ and 5001 and $a = 10$ nm. Despite the small number of realizations, Fig. 2 shows that the model with $N_M = 1001$ leads to $\rho_M(E)$ nearly indistinguishable from that with $N_M = 5001$.

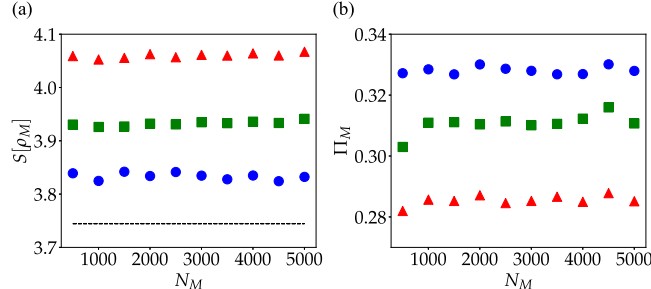

**Fig. 3 Thermodynamic limit convergence of cavity effect on the local molecular density of states entropy and excited-state survival probability.** a Entropy of molecular LDOS as a function of $N_M$ for systems with equal $\Omega_R$ and varying intermolecular distances $a = 10$ nm (blue circles), $a = 25$ nm (green squares) and $a = 50$ nm (red triangles). The dashed line gives the entropy of the bare molecular LDOS. In all computations, $N_M = N_C$, $\sigma/\Omega_R = 0.2$ and $\sigma_\mu = 0.05\mu_0$. Other parameters are given in the text. (b) Molecular return probabilities for the same systems. Each data point corresponds to an average of over 10 realizations.

These observations are quantitatively confirmed in Fig. 3, where we find that when $N_M = N_C > 1001$, both $S[\rho_M]$ and $\Pi_M$ depend only on the molecular density $1/a$. This feature ensures that we have reached the thermodynamic limit for these observables, and justifies our utilization of $N_M = N_C = 1001$ in subsequent sections of this article.

**Microscopic states.** Our computations reveal quasi-extended and localized low-energy polariton states in qualitative agreement with earlier work focused on disorder effects on 1D polaritonic states[55,57] (although, we note that the Hamiltonians used in these studies break time-reversal symmetry and therefore lead to quantitatively distinct properties relative to our work since coherent localization is generally weakened when time-reversal symmetry is absent). These qualitative features of polariton localization in photonic wires have been thoroughly investigated so we provide no detailed discussion, except to mention that there is a clear analogy between the polariton states obtained in photonic wires and the local exciton ground states and the quasi-extended exciton states of polymers[67,68]. This analogy is a consequence of Anderson localization universality according to which, in general, there are no extended states over a 1D system in the thermodynamic limit. Instead, any small amount of disorder leads to a breakdown of long-range order.

In photonic wires strongly coupled to a resonant material, wave function localization, and more generally, the expected distance-dependent decay of intermolecular correlations (mediated by the optical cavity) emerge only when both disorder and multiple electromagnetic field modes are included in the light–matter Hamiltonian.

**Density and energetic disorder dependence.** In Fig. 4, we report the changes induced by strong light–matter interactions in the molecular LDOS entropy (Eq. (6)) and the molecular excited-state escape probability $\chi_M$ (Eq. (10)) as a function of energetic disorder for various mean intermolecular distances at zero detuning and $\Omega_R = 0.3$ eV. Both $\Delta S[\rho_M]$ and $\chi_M$ show a generic decay with increasing disorder (except when $a = 10$ nm and $\sigma/\Omega_R < 0.4$ where $\chi_M$ has a shallow local minimum). This behavior is not surprising: when the molecular system is perfectly ordered, all excitations are maximally delocalized across the entire system (including the weakly coupled molecular states with negligible photonic content) and both $\Delta S[\rho_M]$ and $\chi_M$ take their largest possible value. For even small values of $\sigma/\Omega_R < 0.1$, scattering

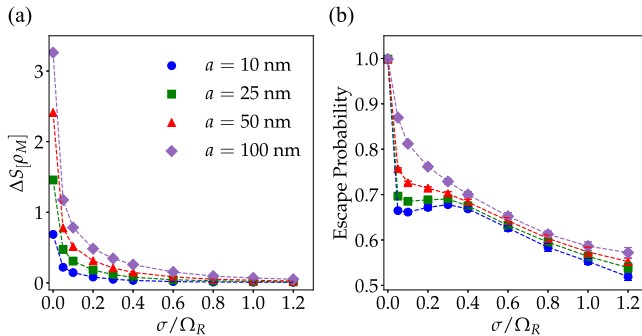

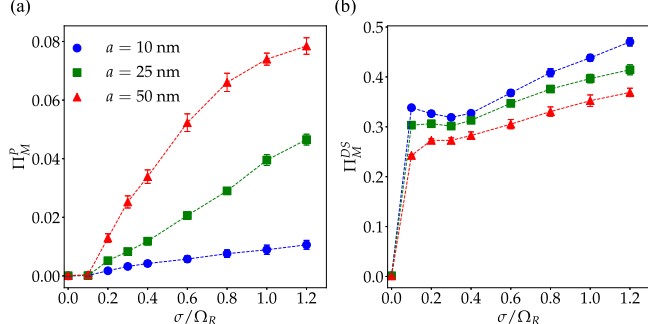

**Fig. 4 Energetic disorder effect on entropy gained by molecular LDOS and escape probability of molecular excited-states. a** Entropy gained by the molecular LDOS upon strong coupling with a photonic wire as a function of $\sigma/\Omega_R$ for varying intermolecular distances (molecular density) with fixed $\Omega_R$ and vanishing detuning $E_C - E_M = 0$. In all computations, $N_M = N_C$, $\sigma_\mu/\mu_0 = 0.05$, and $f = 0.1$. Other parameters are given in the text (see Thermodynamic limit convergence). **b** Molecular excited-state escape probabilities for the same systems. Each point corresponds to an average obtained over 20 disorder realizations (error bars indicate standard deviation).

**Fig. 5 Polariton and dark-state delocalization dependence on the energetic disorder and density for fixed $\Omega_R$. a** Total contribution of polariton states (with photonic content >15% but <85%) to the molecular excited-state survival probability vs. energetic disorder (with zero detuning and $\Omega_R = 0.3$ eV). **b** Total contribution of dark states (with molecular content >85%) to the molecular survival probability. Excited-state localization leads to an increase in the survival probability and therefore, we can infer both polaritons and reservoir modes become more localized with increasing $\sigma$, although the effect is much stronger on the latter, especially at small values of $\sigma/\Omega_R$. Each point corresponds to the average over 20 disorder realizations (error bars indicate standard deviation).

induced by energetic disorder induces weak and strong localization of polaritonic and reservoir modes, respectively. This notion is corroborated by Fig. 5a, b, which show the polariton and dark-state contributions to the exciton survival probability, respectively (according to the criterion that polaritons are hybrid states where the photonic content is greater than 15%, but less than 85%, and reservoir modes have >85% molecular composition—while quantitative results depend on these choices, qualitative features are robust). According to our results, reservoir modes are highly sensitive to disorder, undergoing strong localization even when $\sigma/\Omega_R \leq 0.1$. Conversely, the polaritonic contribution to the survival probability is minimal at small values of $\sigma/\Omega_R$, but becomes significant when the energetic disorder is sufficiently large, further confirming that both weakly and strongly coupled excitations become localized over distances smaller than the size of the system. These trends agree with general ideas of wave function localization theory, namely that exciton localization becomes more prominent, in general, as $\sigma$ increases.

The consistent increase of $\Delta S[\rho_M]$ and $\chi_M$ with the intermolecular distance is more intriguing, since in the thermodynamic limit polariton scattering induced by energetic fluctuations is expected to be independent of $a$ when $q$ is nearly conserved[53]. Moreover, while elastic scattering induced by structural fluctuations is expected to be stronger as $a$ approaches $1/q$, we show numerically in Supplementart Note 2 that unless $\sigma$ vanishes, energetic disorder provides the dominant contribution to polariton and reservoir localization.

The origin of the enhancement of photonic wire effects on the molecular ensemble with $a$ for fixed $\Omega_R$ may be understood by noting that if the cavity modes are integrated out in a path integral representation of the partition function of the system[69,70], the effective action for the molecular system acquires retarded two-body intermolecular interactions mediated by the cavity with coupling constant proportional to the product of the magnitudes of the single-molecule transition dipole moments[33,63,71]. For systems with fixed $N_M$ and $\Omega_R$, the mean transition dipole moment is proportional to $1/\sqrt{\rho} \propto \sqrt{a}$, (since $\Omega_R \propto \sqrt{\rho\mu^2}$ and $\rho \propto 1/a$). Therefore, when $\Omega_R$ and $N_M$ are fixed, the cavity influence on each molecule becomes stronger with decreasing value of density. Evidence for this feature can be obtained from the dark-state contribution to $\Pi_M$ in Fig. 5 (right). This shows the

reservoir modes become significantly more delocalized when $a$ is increased from 10 to 50 nm. Note that, while the polaritonic contribution to $\Pi_M$ in Fig. 5 also increases with $a$, the gain in escape probability is dominated by the reservoir modes since their contribution to $\Pi_M$ is nearly one order of magnitude larger than the polaritonic.

**Cavity detuning effects.** In several chemical reactions where strong influence by optical cavities has been reported, the obtained data suggest that the photonic material maximizes its effect on the reactive system when a specific transition of the reactant is on-resonance with the $q = 0$ mode of the optical cavity[24–26,29]. This feature motivates our study of $S[\rho_M]$ and $\chi_M$ as a function of cavity detuning. In Fig. 6, we present the results given by our multimode theory. The same figure also includes the predictions of theories including a single cavity mode, but a discussion of these results is left to the next section.

As shown by Fig. 6, under strong coupling with a photonic wire, the local molecular DOS entropy and the exciton escape probability are maximally affected by the optical cavity when the lowest-energy photon mode is redshifted relative to the molecular system. In our disordered multimode model, the photonic material is least effective at modifying the low-energy properties of the molecular ensemble when the photons are all blue-shifted relative to the excitons. This trend is robust with respect to changes in the molecular density, and the observed dependence of $\chi_M$ and $S[\rho_M]$ on the density follows the pattern discussed in the previous subsection.

The maximization of the effects of multimode photonic devices on low-energy observable properties (i.e., depending only on the ground and first excited-states of the excitonic subsystem) of the molecular system when $E_C(0) < E_M$ can be understood as follows. First, when the mean molecular excited-state energy is smaller than or equal to all allowed cavity-photon energies ($E_C(0) - E_M \geq 0$), a significant fraction of the molecular excitons will be off-resonant with all cavity modes and the spectral overlap of photonic and molecular modes will be weaker (See bottom of Fig. 1b for a comparison of the case where $E_C(0) = E_M$ and $E_C(0) < E_M$ showing that a large number of molecules will be off-resonant with

all cavity modes when $E_C(0) = E_M$). This notion is confirmed in Fig. 7 (bottom left and right), where we show that the number of polaritons $N_P$ becomes largest at negative detuning, whereas the number of reservoir modes $N_{DS}$ is minimized in redshifted cavities.

Similarly, Fig. 7 (top right) demonstrates that in redshifted cavities, dark states become more delocalized in comparison to cavities with vanishing or positive detuning ($\Pi_M^{DS}$ is smallest for $E_C(0) - E_M < 0$). This feature can also be ascribed to the greater mixing of the reservoir modes with the cavity when the latter is redshifted.

In addition to the reduced light–matter spectral overlap, hybrid modes formed from strongly interacting cavity photons with $q \approx 0$ tend to localize over smaller distances than polaritons dominated by wave-vectors with larger magnitude, since in states with long-distance coherence the wave-vector uncertainty $\delta q$ is much smaller than $q$[16,53,54]. Therefore, in devices where the molecules are only resonant with $q \approx 0$ transitions (zero detuning), any small amount of disorder will lead to polariton modes with finite $\delta q > q$. The latter are localized over an interval with a length smaller than the wavelength, and any delocalization effects on the molecular system induced by light–matter hybridization will be weaker relative to the case where the molecules are resonant with cavity modes with larger $q$[16,54,57]. This effect is easier to notice for the systems with $a = 25$ and 50 nm in Fig. 7. For these intermolecular distances, the number of polaritons decreases significantly when $E_C(0) - E_M$ approaches zero from negative detuning, but the polaritonic contribution $\Pi_M^P$ to the survival probability increases slightly. This confirms that each polariton becomes on average significantly more localized when the cavity is on-resonance with the molecular system at $q = 0$ in comparison to the redshifted case.

Note that, although our numerical results are strictly valid for one-dimensional systems, both arguments employed to explain the observed trends are independent of the dimensionality of the photonic and molecular system. Therefore, the detuning behavior reported in Fig. 6 is expected to hold generically for more realistic treatments of the molecules and the photonic device.

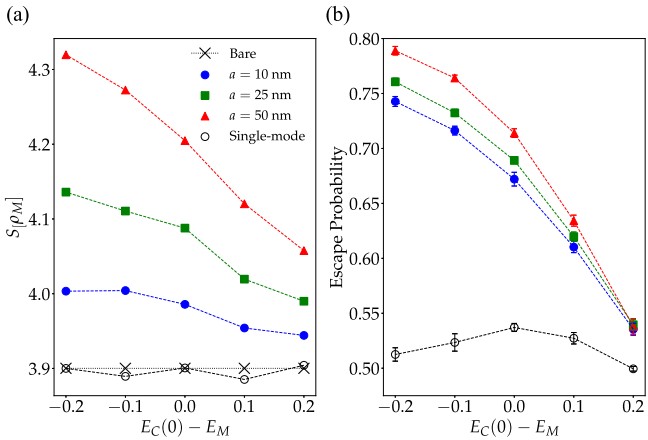

**Fig. 6 Detuning effect on entropy gained by molecular LDOS and escape probability of molecular excited-states. a** Entropy of molecular local density of states as a function of cavity detuning for systems with varying particle densities (but an equal number of molecules $N_M = 1001$) and $\sigma / \Omega_R = 0.2$ (other parameters are as in prior figures). The dashed line corresponds to the entropy of the bare LDOS. **b** Escape probability ($\chi_M = 1 - \Pi_M$) for molecular excitons as a function of cavity detuning for the same model systems. Note the escape probability vanishes for bare molecular states. Each point corresponds to the average over 10 disorder realizations (error bars indicate standard deviation).

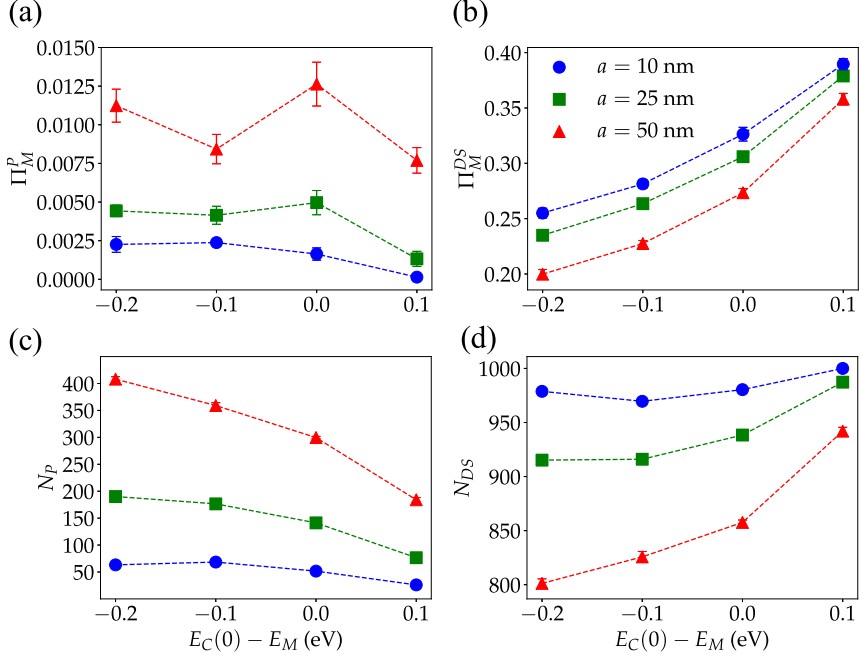

**Fig. 7 Detuning dependence of polariton and dark-state contributions to molecular excited-state survival probability. a** Total contribution of polariton states (states with >15% but <85% photonic content) to the molecular survival probability vs. cavity detuning with $\sigma / \Omega_R = 0.2$ and $\Omega_R = 0.3$ eV. **b** Total contribution of dark states (with greater than 85% molecular content) to the molecular survival probability. **c** Estimated number of polariton states. **d** Estimated number of dark states. Dark states become more numerous and significantly more localized when the cavity-matter detuning goes from negative to positive. In addition, when $a = 25$ and 50 nm, while the number of polariton modes decreases significantly as $E_C(0) - E_M$ goes from $-0.2$ eV to 0 eV, the total polariton contribution to the molecular excited-state survival probability remains roughly the same indicating enhanced polariton delocalization at redshifted cavities. This effect is reduced when $a = 10$ nm. Each point corresponds to the average over 10 disorder realizations (error bars indicate standard deviation).

Our observations do not contradict experimental results on polariton effects on chemical reactions, since we only investigated the influence of strong light–matter interaction effects on low-energy properties of the molecular system involving only the ground and first excited-states of the material, while the energy scales and relevant microscopic states governing thermal chemical reaction rates can be much higher (especially for some of the reactive systems studied experimentally where barrier crossing timescales range from minutes to hours).

Additionally, we assumed that only the lowest-energy cavity-photon band interacts with the molecular system whereas in infrared optical cavities it is typical for the molecular transitions to have $N > 1$ resonances[72]. The zero-detuning condition is then satisfied only by one of the resonant cavity bands, whereas the remaining $N - 1$ have resonances at $q \neq 0$. Further work is required to establish whether the inclusion of multiple polariton bands allows recovery by low-energy effective models of the approximate experimental trend that zero-detuning cavities exert a stronger influence on the local properties of a molecular system.

**Comparison to single-mode theories.** Before concluding, we contrast the main results of this article with the behavior predicted by optical cavity models that include only a single mode.

In systems with equal $\Omega_R$, $\sigma$, and detuning, Fig. 6 shows that single-mode (0D) cavities act on strongly coupled molecular systems in a qualitatively distinct fashion relative to a multimode photonic device. This is unsurprising, since the photonic modes of 0D microcavities are isolated. Therefore, the spectral overlap between the cavity and the molecular subsystem is maximized when they are resonant. This leads to a maximal cavity-mediated exciton escape probability at zero detuning (Fig. 6), since if the retained-mode were off-resonant with the molecular system, the spectral overlap of the material and the cavity would be weaker. This behavior is exclusive to 0D systems with isolated modes and does not apply to cavities with a continuous spectrum, such as those employed in almost all polariton chemistry experiments. In FP cavities or photonic wires, negative cavity detuning gives enhanced light–matter spectral overlap, and a larger number of propagating (delocalized) polariton modes relative to a cavity with zero detuning. Hence, the low-energy properties of molecular ensembles are expected to be more affected by a photonic material when it is redshifted relative to the molecules. A related conclusion was reached in Ref. [31] upon analysis of the viability of a hypothetical mechanism for cavity effects on charge transfer rates[73].

Additional examples of qualitative disagreements between single- and multimode-cavity theories are displayed in Fig. 8, where we compare the cavity-modified $\rho_M(E)$ (blue) and bare $\rho_M^0(E)$ (red) at zero (left panel) and negative (right panel) detuning with equal $\Omega_R$ and $N_M$. The top panel (Fig. 8a, b) contains the results obtained assuming a single electromagnetic mode interacts with the molecular system, whereas the middle (Fig. 8c, d) and bottom (Fig. 8e, f) panels show the predicted $\rho_M(E)$ for multimode cavities where the intermolecular distances are in average equal to 10 nm and 25 nm, respectively.

Figure 8a, b shows that the molecular LDOS is essentially unaffected by a single-mode optical cavity, and $\rho_M(E)$ is indistinguishable from $\rho_M^0(E)$. On the other hand, Figs. 8c–f show the effect of a multimode photonic wire on the molecular LDOS is finite and particularly prominent at the tails of $\rho_M(E)$. This feature leads to finite $\Delta S[\rho_M]$ for a multimode cavity (in contrast to a single-mode theory, for which $\Delta S[\rho_M] = 0$). In addition, the single-mode cavity is completely insensitive to the intermolecular distance, whereas our multimode photonic wire captures the stronger effects of the electromagnetic field on molecular ensembles with greater dipole moment per molecule required to preserve the Rabi splitting

when the molecular density is smaller (see discussion under "Density and energetic disorder dependence").

Contrary to the predictions of single-mode theories[48], we also find that, except for a narrow interval of values of $\sigma/\Omega_R$ where the escape probability does not change too much (Fig. 4), the energetic disorder tends to localize both dark and polaritonic excitations, thus resulting in a reduced escape probability for molecular excited-states.

## Conclusions

We investigated spectral and transport properties of disordered molecular excitons under collective strong coupling with a photonic wire. Our results suggest that low-energy effects of polaritons on bulk properties of the molecular ensemble are largest when the single-molecule transition dipole moment is large, energetic disorder is small, and when the cavity is redshifted (negatively detuned) relative to the molecular system. In negatively detuned devices, there exists enhanced spectral overlap between the cavity and the molecular system. In addition, polariton modes with significant molecular and photonic content have a mean wave-vector far from zero, and delocalization is protected from the small fluctuations induced by structural and energetic fluctuations.

The detuning dependence of cavity effects on molecular properties is particularly relevant when viewed in the context of recent research on polariton effects on thermal chemical reactions[24–27]. Several of the reported experiments imply that photonic materials are more effective at changing reaction rates when the molecular system is resonant with the lowest-energy mode of a cavity band (zero detuning). Our results suggest otherwise but do not contradict experiments, since we only assess low-energy properties of the molecular system, whereas the studied slow chemical reactions are complex events often involving higher excited-states than those included in our model.

Increasing the number of molecules, cavity-photon modes, dimensionality, and introducing dynamical disorder and cavity leakage is unlikely to qualitatively change any of the trends reported in this work. We expect the behavior of molecular observables with cavity detuning, energetic disorder, and density (for a fixed $\Omega_R$), to be generic for low-energy polariton models. For example, while wave function delocalization is significantly more favored in 2D and 3D[74] relative to the case studied in this paper, it remains true that the spectral overlap between excitons and the optical cavity will be weaker at zero or positive detuning when compared to negative. Dynamical disorder (time-dependent fluctuations of the exciton frequency, position, and dipole magnitude/orientation) induces exciton localization and ultrafast coherence decay[75] further reducing the efficiency of inter-molecular exciton transport reported here, but is unlikely to affect any of the qualitative trends reported.

To conclude, we reiterate that although 0D single-mode cavity theories are insightful and predictive of many features of polaritonic optical response, these models give incorrect qualitative predictions for the cavity effects on molecular ensembles analyzed in this work. Specifically, single-mode models fail to capture the detuning, density, and disorder strength dependence of the exciton escape probability and molecular LDOS entropy gain upon collective strong coupling with a photonic device. These shortcomings must be recognized in future model building aimed at predicting novel ways to control chemistry with optical microcavities.

## Methods

Exact diagonalization was performed for multiple realizations of the random light–matter Hamiltonian in Eq. (3), and the eigenstates of each realization were employed to compute averages of the molecular ensemble quantities $\rho_M(E)$, $S[\rho_M]$, and $\Pi_M$ as described in the main text. The criteria used to identify states as

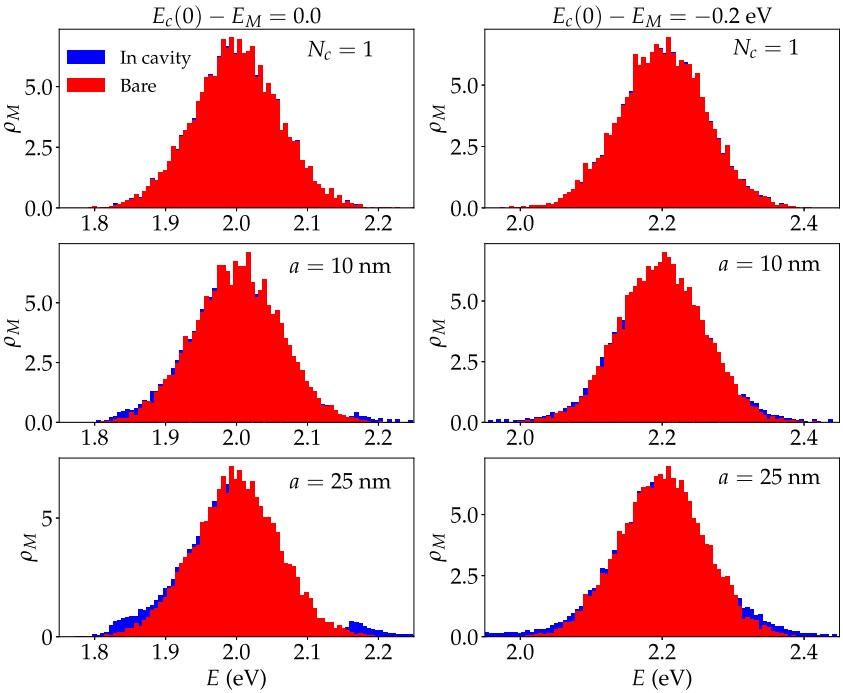

**Fig. 8 Polariton effects on molecular LDOS of single and multimode cavities.** Local molecular density of states for 10 realizations of a molecular system in free space (red) and under strong coupling (blue) to a photonic wire. $N_M = 1001$, $\Omega_R = 0.3$ eV, $\sigma/\Omega_R = 0.2$, $\sigma_\mu = 0.05\mu_0$, and $f = 0.1$ in all of the studied scenarios. In panels, where only $a$ is specified, $N_C = N_M$. Note that larger single-molecule transition dipole moments are required to preserve $\Omega_R$ when $a$ is increased. This explains the stronger cavity effects on the molecular density of states when $a = 25$ nm).

polaritonic or weakly coupled (dark) states is discussed under Results and Discussion and specified in the captions of Figs. 5 and 7.

## Data availability
The data that support the findings of this study are available from the corresponding author upon reasonable request.

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

## Acknowledgements
R.F.R. acknowledges generous start-up funds from the Emory University Department of Chemistry.

## Competing interests
The author declares no competing interests.
