## [Peer Review File · Communications Chemistry]

Reviewers' comments:

Reviewer #1 (Remarks to the Author):

In this paper by Raphael F. Ribeiro, the spectral and transport properties of a disordered molecular material coupled to a photonic wire are studied. While the results are interesting and timely, I believe that the paper suffers from important loopholes that must be fixed before recommending the manuscript for publication.

-The main claim of the paper is that some results (e.g. the escape probability decreases as the Rabi splitting is increased, and the largest changes are obtained when the cavity is redshifted relative to the molecular absorption) are not captured with theories that only include a single cavity mode. I find it surprising that this somehow interesting claim is not supported by any data in the paper. I would thus recommend including a direct comparison between the results obtained for a single-mode and a multimode cavity.

-While disorder on the transition dipole moment is included (in addition to fluctuations in the transition frequencies), its sole effect on the spectral and transport properties is not discussed at all. Is this type of disorder important in the results? Would anything change by setting it to zero?

-Why is the regime where the variance is larger than the Rabi splitting not discussed at all?

-Wouldn't it be possible to study the contribution of the polaritons and dark reservoir to the observables shown in Fig. 4 and 5. I would expect this type of analysis particularly helpful, in particular to understand the differences with the results of [44] and [54].

-I did not understand the claim in line 345 that the gain in entropy is essentially zero in the single-mode case.

-I did not find the explanation of why the observed effects are maximum when the cavity is redshifted relative to the molecular absorption particularly useful. As it is based on the results presented in two other papers [46-47], this discussion should be expanded and reformulated. Perhaps a sketch could help improving clarity here.

-Fig. 1 could also be improved. Perhaps some labels like L and a could be added to (a). It also seems to me that the color code in (c) is not correct.

Reviewer #2 (Remarks to the Author):

The manuscript by Raphael Ribeiro focuses on the role of the multi-mode structure of the confined electromagnetic field in optical microcavities on the delocalization properties of electronic excited states in a disordered molecular array. The approach is a welcome extension of early studies in the 2000's (Litinskaya et al.) and more recent work on this topic by Feist, Schachenmayer, Scholes and others. These previous works are fairly mentioned in the initial review sections of the manuscript. Using a linear array coupled with 1D waveguide field as a numerically tractable example, the author studies the eigenstates and spectrum of the coupled light-matter system, taking into account Gaussian static energy disorder of the molecular levels in the array, uniform static fluctuations of the

molecular positions, and static fluctuations of the dipole moment magnitudes. Strong coupling is assumed at the bottom of the photon dispersion curve.

One of the interesting insights from the calculations is the dependence of the delocalization properties on the lattice constant, which is not present in the most intensely studied Dicke regime. It is not clear however how these findings relate the numerical analysis in ref. 48 and the analytical work in ref. 46, where the degree of polariton coherence is also related to the density of states in momentum space, thus by the lattice constant at fixed array length.

The other interesting point that is not very clearly developed is the entropy of the density of states in eq. 6. This is arguably not a usual way to characterize state delocalization, and given its pivotal role in the analysis one would expect a more comprehensive exposition of its properties for limiting cases (e.g., Anderson limit, disorder-free, single-molecule, etc.), so one can better assess the validity of statements in lines 262-264 (Π_M measures energy diffusion, and $\rho_M(E)$ directly relates to thermodynamic and kinetic quantities) and also better understand the physical meaning of figs. 4 and 5, in the context of previous works (e.g, ref. 46)

Figure. 2 would also strongly benefit from a display of the corresponding cavity transmission spectrum, so readers, specially experimentalists, can better appreciate the connection between homogeneous and inhomogeneous linewidths in linear response and how that relates to the degree of localization of the underlying light-matter states.

There are also other minor proofreading issues that need to be corrected.

Publication in Communications Chemistry is recommended after these points are addressed.

Reviewer #3 (Remarks to the Author):

In this manuscript the interaction between a line of two level atoms strongly coupled to an effectively one dimensional optical cavity is modelled in detail. The focus is on incorporating various types of disorder related to the cavity and the molecules that are bound to be present in realistic scenarios. Many models consider only a single mode of the cavity. Here the lowest band of the one dimensional cavity with a spread of wave-vectors and corresponding frequencies along the loosely confined dimension is considered. Scatter in the mean transition energies and positions of the two level atoms/molecules as well as in their respective dipole moments is also incorporated. These three quantities are sampled from suitably chosen distributions.

Numerical simulations of the model are done with choice of parameters that approach the thermodynamic limit. In this case, the key parameters are the number of molecules and the number of cavity modes considered. Both are taken to be equal and numerical investigations show that choosing both around a 1000 gives stable results for the quantities being studied with negligible further variation when the numbers are increased further. The main observables that are studied include the molecular local density of states and the probability of escape of a molecule from its

excited state as a function of various parameters related to both molecular and cavity properties. Entropy is the single parameter that is studied in relation to the local density of states and in particular its deviation from the case with no cavity is explored. The presence of disorder in the system leads to the polariton states being more localised in relation to the case with a single sharp mode for the cavity. Important findings from the numerical studies of the model include the following. It is found that with all noise parameters kept constant, the effects of polaritonic states formed due to the strong coupling between the cavity and the molecules are more prominent when the molecular density is low (spacing between molecules is larger). Similarly when there is disorder, red-shifting the cavity relative to the mean (resonant) frequency of the molecules lead to more prominent effects that are attributable to polaritonic states. The limitations of the model in that it considers only two levels of the atom and one band of the cavity are discussed in relation to available experimental results and prospects of the trends observed in the model translating to higher dimensional cases and wider energy ranges are also discussed.

This is a well written manuscript with interesting numerical results that can potentially have a bearing on designing experiments on modulating molecular and reaction properties using cavities. While the general problem of modelling such a system is quite hard, this work offers a limited case model incorporating some of the expected characteristics of a realistic cavity-molecule system. The limitations of the simplified model are laid out clearly thereby allowing the reader to judge easily whether the results are relevant to a proposed or already completed experiment. In view of this I am happy to recommend this manuscript as is for publication in Communication Chemistry from Nature press.

Referee response for Strong light-matter interaction effects on molecular ensembles

Raphael F. Ribeiro¹

¹*Department of Chemistry, Emory University*
(Dated: December 28, 2021)

First of all, I want to thank the referees for the insightful comments and pertinent suggestions and questions. Below, I address each. Text from the referee's is given under quotation marks. My response to each issue is given below in blue, and changes to the text are given in italics (some changes of the main manuscript are not explicitly given here but a reference to a specific part of the text is always provided.)

Reviewer #1.

“In this paper by Raphael F. Ribeiro, the spectral and transport properties of a disordered molecular material coupled to a photonic wire are studied. While the results are interesting and timely, I believe that the paper suffers from important loopholes that must be fixed before recommending the manuscript for publication.”

I would like to thank the referee for the attention given to the paper and for considering it to be interesting and timely. The issues raised are addressed below.

- “The main claim of the paper is that some results (e.g. the escape probability decreases as the Rabi splitting is increased, and the largest changes are obtained when the cavity is redshifted relative to the molecular absorption) are not captured with theories that only include a single cavity mode. I find it surprising that this somehow interesting claim is not supported by any data in the paper. I would thus recommend including a direct comparison between the results obtained for a single-mode and a multimode cavity.”

The direct comparison between the results obtained for a single-mode and a multimode cavity was presented in Fig. 5 of the originally submitted manuscript. Nevertheless, in order to further emphasize the shortcomings of single-mode theories (when applied to evaluating cavity effects on molecular ensembles strongly coupled to a multimode cavity), we have reorganized our article by introducing a new heading **Comparison to single-cavity mode theories** (p. 7) under **Results and Discussion**, focused entirely on the comparison between predictions of single-mode and multimode theories such as ours. In addition, I also inserted a figure that compares the predictions given by single and multimode cavity models for the local molecular density of states (Fig. 8 of most recent manuscript). This also helps to show that single-mode theories tend to significantly underestimate the effects of multimode light-matter strong coupling (and further addresses a point raised by the referee below).

- “While disorder on the transition dipole moment is included (in addition to fluctuations in the transition frequencies), its sole effect on the spectral and transport properties is not discussed at all. Is this type of disorder important in the results ? Would anything change by setting it to zero ?”

Transition dipole moment (dipolar) disorder was included to model structural heterogeneity. This type of disorder becomes essential when the σ is infinitesimal relative to Ω_R . For example, in the most extreme case where energetic disorder vanishes, dipolar disorder also drives localization of polaritonic and dark excitations. I added a detailed analysis and comparison of the

structural and energetic disorder effects on the spectral and transport properties of the system studied to the Supplementary Information (Sec. 1). I have also added the following sentence to page 3:

Nevertheless, for the studied model, except for infinitesimal values of σ and simultaneous σ_μ/μ_0 close to or greater than 1, energetic disorder plays a more important role, and therefore, we take σ_μ and f to be constant throughout this work. A quantitative comparison and discussion of energetic and structural disorder effects on molecular observables is given in the Supplementary Material.

- “Why is the regime where the variance is larger than the Rabi splitting not discussed at all ?”

The reason I did not discuss the scenario where the variance is larger than the Rabi splitting is a preference to focus the discussion on the cases that are customarily being probed in experiments where energetic disorder is smaller than the Rabi splitting. Nevertheless, I agree the case raised by the referee is an interesting one, and the new Figs. 4 and 5 include results for $\sigma \geq \Omega_R$. These figures confirm that the molecular escape probability and $\Delta S[\rho_M]$ decay with σ/Ω_R except for a small interval around $0.1 < \sigma/\Omega_R < 0.3$ where an increase in σ leads to tiny changes in the molecular escape probability.

- “Wouldn’t it be possible to study the contribution of the polaritons and dark reservoir to the observables shown in Fig. 4 and 5. I would expect this type of analysis particularly helpful, in particular to understand the differences with the results of [44] and [54].”

I added two new figures (5 and 7) providing a decomposition of the molecular excited-state return probability into disjoint contributions from polaritons and dark states (according to a definition specified now in the last paragraph of page 4 left column). This data was very helpful in providing additional evidence for the analysis of the results we reported in the previous version of our manuscript.

- “I did not understand the claim in line 345 that the gain in entropy is essentially zero in the single-mode case.”

I have reformulated and moved the text related to the entropy of the local molecular density of states predicted by the single-mode case. The following text can now be found in the last paragraph of p. 7:

Density: Additional examples of qualitative disagreements between single- and multimode-cavity theories are displayed in Fig. 8, where we compare the cavity-modified $\rho_M(E)$ (red) and bare $\rho_M^0(E)$ (blue) at zero (left panel) and negative (right panel) detuning with equal Ω_R and N_M . The top panel (Figs. 8a and 8b) contains the results obtained assuming a single electromagnetic mode interacts with the molecular system, whereas the middle (Figs. 8c and 8d) and bottom (Figs. 8e and 8f) panels show the predicted $\rho_M(E)$ for multimode cavities where the intermolecular distances are in average equal to 10 nm and 25 nm, respectively.

*Figures 8a and 8b show that the molecular LDOS is essentially unaffected by a single-mode optical cavity, and $\rho_M(E)$ is indistinguishable from $\rho_M^0(E)$. On the other hand, Figs. 8c–f show the effect of a multimode photonic wire on the molecular LDOS is finite and particularly prominent at the tails of $\rho_M(E)$. This feature leads to finite $\Delta S[\rho_M]$ for a multimode cavity (in contrast to a single-mode theory, for which $\Delta S[\rho_M] = 0$). In addition, the single-mode cavity is completely insensitive to the intermolecular distance, whereas our multimode photonic wire captures the stronger effects of the electromagnetic field on molecular ensembles with greater dipole moment per molecule required to preserve the Rabi splitting when the molecular density is smaller (see discussion under **Density and energetic disorder dependence**.)*

- “I did not find the explanation of why the observed effects are maximum when the cavity is redshifted relative to the molecular absorption particularly useful. As it is based on the results presented in two other papers [46-47], this discussion should be expanded and reformulated. Perhaps a sketch could help improving clarity here.”

I have reformulated Fig. 1 to help the reader understand the essential idea that the cavity-matter spectral overlap (together with the maximization in number of coherent polariton states shown in Refs. 46 and 47 to emerge in redshifted cavities) play a dominant role in the influence of optical cavities on the selected molecular ensemble observables. In particular, it is now explicitly shown that in sufficiently redshifted cavities, there are essentially no molecules off-resonant with the cavity, whereas in cavities with zero detuning, the subset of molecules with excitation energy lower than the average is off-resonant with all cavity modes, and therefore the effects of the cavity on such molecules is smaller. As a result, the effect of the cavity on the examined properties of molecular ensemble is stronger when the cavity is redshifted relative to the molecular system.

The new text added to the caption of Fig. 1 summarizes the above point. The decomposition of the excitation survival probability into polariton and dark state contributions in Fig. 7 as a function of detuning and analysis included in p.7 also provide additional insight.

- “Fig. 1 could also be improved. Perhaps some labels like L and a could be added to (a). It also seems to me that the color code in (c) is not correct.”

Fig. 1a was improved by introducing labels associated to the notation mentioned in the text and as suggested by the referee. The color code in (c) was correct, but in order to minimize the risk of ambiguity I reformulated the presentation of the obtained data to ensure clarity.

I would like to thank referee #1 again for reviewing this manuscript and for the various insightful suggestions and comments.

Reviewer #2.

“The manuscript by Raphel Ribeiro focuses on the role of the multi-mode structure of the confined electromagnetic field in optical microcavities on the delocalization properties of electronic excited states in a disordered molecular array. The approach is a welcome extension of early studies in the 2000’s (Litinskaya et al.) and more recent work on this topic by Feist, Schachenmayer, Scholes and others. These previous works are fairly mentioned in the initial review sections of the manuscript. Using a linear array coupled with 1D waveguide field as a numerically tractable example, the author studies the eigenstates and spectrum of the coupled light-matter system, taking into account Gaussian static energy disorder of the molecular levels in the array, uniform static fluctuations of the molecular positions, and static fluctuations of the dipole moment magnitudes. Strong coupling is assumed at the bottom of the photon dispersion curve.”

I want to thank the referee for their careful reading of this manuscript. Answers to each raised point are given below.

- “One of the interesting insights from the calculations is the dependence of the delocalization properties on the lattice constant, which is not present in the most intensely studied Dicke regime. It is not clear however how these findings relate the numerical analysis in ref. 48 and the analytical work in ref. 46, where the degree of polariton coherence is also related to the density of states in momentum space, thus by the lattice constant at fixed array length. ”

A difference between our work and both mentioned references is that their focus was on polariton coherence measured by the fractional wave-vector uncertainty $\delta q/q$, whereas our manuscript

is focused on the cavity strong coupling effect on selected molecular ensemble observables. Because most eigenstates with significant molecular content are weakly coupled, the ensemble observables discussed in this manuscript are only weakly sensitive to the polariton coherence. Nevertheless, Fig. 1 shows that our simulations are shown to be valid in the thermodynamic limit for all of the a values studied, and therefore, for any value of a , there is no dependence on the length L of the system (Fig. 1), or equivalently on the cavity wave-vector spacing $\Delta q = 2\pi/L \rightarrow 0$ (we are essentially working at the continuum limit). This allows us to compare directly the results obtained with distinct values of a without being concerned about finite-size effects associated to L .

Given that our focus is on observables associated to the molecular subsystem, polariton coherence is expected to only depend weakly on a , since the wave-vector uncertainty in the perturbative limit behaves as $\delta q = \delta E(q)/\hbar\partial\omega(q)/\partial q$ and $\delta E(q)$ is the imaginary part of a polariton with mean wave-vector q originating from its interaction with the fluctuations of the medium. Litinskaya and Reineker (Physical Review B, 74(16), 165320) showed that for an eigenstate with average wave-vector q and energy $E(q)$, energetic disorder leads to the following dispersion equation

$$[E(q) - E_C(q)] \approx \Omega_R^2 \int_{-\infty}^{\infty} dE_j \frac{\rho(E_j)}{E(q) - E_j}, \quad (1)$$

where E_j is the j th molecule excited-state energy and ρ is the molecular excited-state energy distribution. This Eq. contains no dependence on a . The same authors showed that in the presence of dipolar disorder, the elastic scattering mean free path decreases with an increase in a . However, as we show in the SI, structural disorder only plays an important role in our work when the energetic disorder is nearly zero. For these reasons, we argue in our article that the fundamental effect of increasing a with Ω_R fixed is the enhancement of effective intermolecular interactions which can be understood by the fact that $\Omega_R \propto \sqrt{\mu^2/a}$, and therefore, an increase in a is necessarily followed by a proportional increase of $\sqrt{\mu}$. This feature affects primarily delocalization of reservoir modes which tend to dominate the excitations with significant molecular character in the optical cavity, and more importantly, are much more localized than the polariton states, as shown in the new Figs. 5 and 7 of the main text. These points are summarized on the right column (second and third paragraphs) of page 5 of the new manuscript.

- “The other interesting point that is not very clearly developed is the entropy of the density of states in eq. 6. This is arguably not a usual way to characterize state delocalization, and given its pivotal role in the analysis one would expect a more comprehensive exposition of its properties for limiting cases (e.g., Anderson limit, disorder-free, single-molecule, etc.), so one can better assess the validity of statements in lines 262-264 (Π_M measures energy diffusion, and $\rho_M(E)$ directly relates to thermodynamic and kinetic quantities) and also better understand the physical meaning of figs. 4 and 5, in the context of previous works (e.g, ref. 46)”

We thank the referee for raising this point. Indeed, $S[\rho_M(E)]$ is an unusual way to measure delocalization, and we only introduce it here because it allows us to “give a number” to the change induced by the cavity on the local molecular density of states. The physical interpretation of $S[\rho_M(E)]$ is that it tracks delocalization of the molecular excited-states in energy space (as opposed to delocalization in real space which we measure with $1 - \Pi_M$). We have expanded our text to provide a more detailed description of $S[\rho_M(E)]$ and how it informs our analysis. In particular, we reorganized our discussion and added the following two paragraphs to page 4: *$\Delta S[\rho_M]$ allows us to quantify the cavity effect on the molecular ensemble energy fluctuations. Roughly speaking, $\Delta S[\rho_M]$ provides a measure of molecular excited-state delocalization in energy*

space. Therefore, we expect $\Delta S[\rho_M]$ to be a nondecreasing function of Ω_R that is greater than or equal to zero, since polaritons have energies separated from the bare molecule mean excited-state energy by approximately $\pm\Omega_R/2$ (Fig. 1). However, the reduced bare photonic DOS relative to the molecular at energies where the molecular DOS is maximal suggests weakly coupled reservoir states dominate the molecular LDOS and the cavity-driven change in $S[\rho_M]$ is expected to be small.

Π_M provides a measure of excited-state delocalization in real-space and coherent energy diffusion efficiency: in systems where all Hamiltonian eigenstates are delocalized $P_{n\psi} \propto 1/N_M$ for all ψ and n , and therefore, Π_M vanishes in the thermodynamic limit, whereas in noninteracting systems with maximally localized excited-states, each molecule corresponds to a Hamiltonian eigenstate, and therefore $P_{n\psi}(t) = \delta_{n\psi}$ and $\Pi_M = 1$. In our model, the case where $\Pi_M = 1$ corresponds to the molecules outside of the cavity (since we assume direct intermolecular interactions are insignificant), while we find $\Pi_M \rightarrow 0$ when energetic disorder vanishes (see Fig. 4). Dark states are expected to have much large contribution to Π_M than polariton modes since the latter inherit greater delocalization from their strong mixing with cavity modes.

- “Figure. 2 would also strongly benefit from a display of the corresponding cavity transmission spectrum, so readers, specially experimentalists, can better appreciate the connection between homogeneous and inhomogeneous linewidths in linear response and how that relates to the degree of localization of the underlying light-matter states. ”

Although we agree with the referee that the connection between inhomogeneous and homogeneous molecular linewidths is of importance to the interpretation of polariton transmission spectra, our work is focused on the effects of the optical cavity and strong light-matter interactions on spectral and (coherent) transport properties of the molecular ensemble given the significant interest in uncovering why certain intrinsically chemical phenomena such as reaction rates and equilibrium constants have shown to be sometimes drastically impacted by optical cavities. The effects of disorder on optical response properties are important and we have ongoing work on related issues, but they fall outside the scope of the current manuscript, which is focused on coherent transport and spectral fluctuations.

- “There are also other minor proofreading issues that need to be corrected. ”

We have fixed typos that were found during this review process.

“Publication in Communications Chemistry is recommended after these points are addressed.”

We are thankful to the referee for the very thoughtful feedback and suggestions of issues to address/clarify.

Reviewer #3.

“In this manuscript the interaction between a line of two level atoms strongly coupled to an effectively one dimensional optical cavity is modelled in detail. The focus is on incorporating various types of disorder related to the cavity and the molecules that are bound to be present in realistic scenarios. Many models consider only a single mode of the cavity. Here the lowest band of the one dimensional cavity with a spread of wave-vectors and corresponding frequencies along the loosely confined dimension is considered. Scatter in the mean transition energies and positions of the two level atoms/molecules as well as in their respective dipole moments is also incorporated. These three quantities are sampled from suitably chosen distributions.”

“ Numerical simulations of the model are done with choice of parameters that approach the thermodynamic limit. In this case, the key parameters are the number of molecules and the number of cavity modes considered. Both are taken to be equal and numerical investigations show that choosing both around a 1000 gives stable results for the quantities being studied with negligible further variation when the numbers are increased further. The main observables that are studied include the molecular local density of states and the probability of escape of a molecule from its excited state as a function of various parameters related to both molecular and cavity properties. Entropy is the single parameter that is studied in relation to the local density of states and in particular its deviation from the case with no cavity is explored. The presence of disorder in the system leads to the polariton states being more localised in relation to the case with a single sharp mode for the cavity.”

“Important findings from the numerical studies of the model include the following. It is found that with all noise parameters kept constant, the effects of polaritonic states formed due to the strong coupling between the cavity and the molecules are more prominent when the molecular density is low (spacing between molecules is larger). Similarly when there is disorder, red-shifting the cavity relative to the mean (resonant) frequency of the molecules lead to more prominent effects that are attributable to polaritonic states. The limitations of the model in that it considers only two levels of the atom and one band of the cavity are discussed in relation to available experimental results and prospects of the trends observed in the model translating to higher dimensional cases and wider energy ranges are also discussed. ”

“This is a well written manuscript with interesting numerical results that can potentially have a bearing on designing experiments on modulating molecular and reaction properties using cavities. While the general problem of modeling such a system is quite hard, this work offers a limited case model incorporating some of the expected characteristics of a realistic cavity-molecule system. The limitations of the simplified model are laid out clearly thereby allowing the reader to judge easily whether the results are relevant to a proposed or already completed experiment. In view of this I am happy to recommend this manuscript as is for publication in Communication Chemistry from Nature press.”

We want to thank the referee for their careful reviewing of our manuscript and for their appreciative comments.

Reviewers' comments:

Reviewer #1 (Remarks to the Author):

The author has fully and satisfactorily addressed my questions/comments. I believe the manuscript has been significantly improved and I thus recommend publication as is.

Reviewer #2 (Remarks to the Author):

The author has convincingly clarified the use of a state entropy measure of delocalization, which was one of the main questions in my original report. This is a very interesting proposal will surely stimulate further theoretical work. Further clarifications and extended discussions in response to another reviewer have also improved the manuscript.

Despite these improvements, the comparison with earlier work in refs. 46, 48 is not yet convincing. The question was not about finite-size effects, but about the dependence of the density of states on the lattice constant in the thermodynamic limit. An explicit comparison between ref. 48 and this work should be brought to the main text, so expert readers can better judge the conceptual consistency of the numerics in this manuscript. Simply presenting in the response without changes to the text is not sufficient.

Regarding my comment on Fig. 2, as a minimum we should know where the LP and UP transmission peaks are located in the energy scales (e.g., with arrow labels, etc.) . Spectral measures are always pivotal discussion points even when discussing transport properties only.

After improving the response to my original comments along the lines mentioned above, the manuscript can be recommended for publication in Communications Chemistry.

Referee response for Strong light-matter interaction effects on molecular ensembles

Raphael F. Ribeiro¹

¹*Department of Chemistry, Emory University*
(Dated: January 27, 2022)

I want to thank the referees again for taking another look at our manuscript. Below I address the two points raised by referee #2 given that referee #1 recommended acceptance without further changes.

The referee’s comments are given under quotation marks. My response to each issue is given in blue, and changes to the text are given in italics (some changes to the main manuscript are not explicitly given here but a reference to a specific part of the text is always provided.)

Reviewer #2.

“The author has convincingly clarified the use of a state entropy measure of delocalization, which was one of the main questions in my original report. This is a very interesting proposal will surely stimulate further theoretical work. Further clarifications and extended discussions in response to another reviewer have also improved the manuscript.”

Despite these improvements, the comparison with earlier work in refs. 46, 48 is not yet convincing. The question was not about finite-size effects, but about the dependence of the density of states on the lattice constant in the thermodynamic limit. An explicit comparison between ref. 48 and this work should be brought to the main text, so expert readers can better judge the conceptual consistency of the numerics in this manuscript. Simply presenting in the response without changes to the text is not sufficient.

I want to thank the referee for the clarification. A discussion of this issue was introduced to the main text on p. 3:

In later sections, we characterize the dependence of local molecular observables on energetic disorder, cavity detuning, and mean intermolecular distance (or photonic lattice constant) for fixed $N_M = N_C$ and Ω_R . The study of lattice constant effects with fixed $N_M = N_C$ is a distinctive feature of our work in relationship to Refs. [48, 50] (who analyzed polariton coherence in the thermodynamic limit with fixed a) and is motivated by the following question: are there significant differences in the polariton effects on molecules showing equal Ω_R but different molecular densities ($1/a$ in our photonic wire)? Note that a change in the lattice constant a to $f \times a$ ($f > 0$) leaves Ω_R fixed under the mentioned conditions if and only if $\mu_0 \mapsto \sqrt{f}\mu_0$ (as $\Omega_R \propto \mu_0 a^{-1/2}$), as expected since a reduced molecular density requires greater transition dipole moment per molecule to preserve a given Rabi splitting. Nonetheless, a potential unintended consequence of changing a with fixed N_C in cavity lattice models is that the photon density of states $\rho_C(\omega) = \sum_q \delta(\omega - \omega_q)$ is also modified. In the thermodynamic limit, in fact, $\rho_C(\omega) \rightarrow [L/(2\pi)] \int_{\mathbb{R}} dq \delta(\omega - \omega_q)$ is proportional to L , and only $\rho_C(\omega)/L$ is independent of L . In Fig. 1, we show that the effects of optical microcavities in the molecular ensemble measured in our study are essentially independent of L as long as sufficiently large number of modes is introduced in the theory. This suggests that the change in photon density of states that results from varying a (with fixed $N_C = N_M$ and Ω_R) is immaterial to our conclusions. In the Supplementary Material, we provide numerical evidence that further supports these points.

We also added a section to the Supplementary Information where we show that using a fixed length L (and thus enforcing equal photon density of states for the cavities with different values of a) in cases with different lattice constants does not give results with relevant differences from the computations in the main manuscript where L depends on the lattice constant, thus, indicating that the lattice constant effect on the photon density of states can be ignored for our purposes.

“Regarding my comment on Fig. 2, as a minimum we should know where the LP and UP transmission peaks are located in the energy scales (e.g., with arrow labels, etc.) . Spectral measures are always pivotal discussion points even when discussing transport properties only.”

Figure 1 was changed in a manner that addresses the referee’s comments. In particular, the energies for LP and UP modes are now included. We also added a color bar and scale which allows one to identify polaritonic, nearly purely photonic and molecular weakly coupled modes via the photonic content of each excitation.

“After improving the response to my original comments along the lines mentioned above, the manuscript can be recommended for publication in Communications Chemistry.”

I want to thank the referee again for his time and careful consideration of my manuscript and its revision, and hope that I have now fully addressed the raised issues.

REVIEWERS' COMMENTS:

Reviewer #2 (Remarks to the Author):

The remaining points of the previous report have been sufficiently addressed in the updated manuscript. Publication can be recommended.